# Novel Unconventional-Active-Jamming Recognition Method for Wideband Radars Based on Visibility Graphs

**DOI:** 10.3390/s19102344

**Published:** 2019-05-21

**Authors:** Congju Du, Bin Tang

**Affiliations:** School of Information and Communication Engineering, University of Electronic Science and Technology of China, Chengdu 611731, China; bint@uestc.edu.cn

**Keywords:** radar unconventional active jamming, jamming recognition, visibility graphs, feature extraction, random forests

## Abstract

Radar unconventional active jamming, including unconventional deceptive jamming and barrage jamming, poses a serious threat to wideband radars. This paper proposes an unconventional-active-jamming recognition method for wideband radar. In this method, the visibility algorithm of converting the radar time series into graphs, called visibility graphs, is first given. Then, the visibility graph of the linear-frequency-modulation (LFM) signal is proved to be a regular graph, and the rationality of extracting features on visibility graphs is theoretically explained. Therefore, four features on visibility graphs, average degree, average clustering coefficient, Newman assortativity coefficient, and normalized network-structure entropy, are extracted from visibility graphs. Finally, a random-forests (RF) classifier is chosen for unconventional-active-jamming recognition. Experiment results show that recognition probability was over 90% when the jamming-to-noise ratio (JNR) was above 0 dB.

## 1. Introduction

Electronic countermeasure (ECM) techniques, such as active deceptive jamming, can generate false targets imitating real ones, and even disturb target detection or tracking [1,2]. Active barrage jamming is another ECM technique, which uses noise or noise-like signal to barrage the target echo, preventing the radar from detecting targets or measuring target parameters. Its basic principle is to use high-power random or pseudo-random signals, thereby reducing radar-detection probability. On the other hand, with the rapid development of radar electronic counter-countermeasure (ECCM) techniques, ECCM techniques, including side-lobe cancellation and blanking [3], space-time adaptive processing [4] and adaptive beamforming [5], have gradually weakened the jamming performance of radar conventional active jamming.

Unconventional active jamming is generated by replicating the transmitted signals of the victim radar, followed by a series of parameter modulations based on digital-radio-frequency memory (DRFM) technology. Compare with conventional active deceptive and barrage jamming, it is coherent with the radar transmitted signals and can achieve some processing gain from signal processing that includes pulse compression and coherent accumulation. Therefore, conventional ECCM techniques become ineffective, and the jamming always enters the radar receiver. One of the most effective schemes to weaken the jamming is the so-called signal-processing based scheme, which is designed to recognize and suppress jamming based on the difference between the echo and the jamming. Commonly used ways to recognize radar active jamming can be divided into two steps, feature extraction and classifier design. The distinguishability of jamming-signal features directly affects the recognition probability of subsequent classifier design. The majority of feature extraction operates on the time/frequency/spatial [6], polarization [7], statistical [8,9], and multiscale-joint domains [10]. These features show the information of the jamming from different perspectives, and we consider a new perspective to extract jamming features in this paper.

Time series analysis is the use of statistical methods to analyze time series data and extract meaningful statistics and characteristics of the data. Therefore, nonlinear time-series analysis can be characterized by the complexity in the signal owing to jamming signals always carrying some fingerprint features [11,12]. At the end of the 19th century, with the rapid development of computer and Internet technology, the theory of complex networks [13,14] was proposed to outline the real-world network. Complex-network theory could be a useful way to characterize time series, and a series of related achievements, including visibility graphs and horizontal visibility graphs [15,16], recurrence networks [17,18], and correlation networks [19], were presented for analyzing the structural properties of time series by complex networks. The visibility algorithm was originally applied to computational geometry and robot motion planning. In 2008, Reference [15] used the visibility algorithm for time-series analysis by converting time series into graphs, called visibility graphs. Up to now, the visibility algorithm has been used in many fields, such as financial [20], traffic [21], and geographic time series analysis [22].

In this paper, we propose a novel method for radar unconventional-active-jamming recognition based on visibility graphs. The representations for different types of unconventional active jamming were derived for illustration purposes in Section 2. In Section 3, the visibility algorithm that converts a time series into a graph is given and the concept of visibility graphs is introduced; then the feasibility of the visibility algorithm for radar signals is analyzed. Section 4 extracts four features on visibility graphs that are valid for jamming recognition. Then, Section 5 outlines numerical results and analysis for unconventional-active-jamming recognition based on a random-forests classifier. Finally, in Section 6, the main conclusions of the paper are summarized.

## 2. Characteristics of Unconventional Active Jamming

### 2.1. Unconventional Deceptive Jamming

At present, most unconventional deceptive jamming uses DRFM technology to achieve accurate the interception of radar signals to generate various types of deceptive jamming. These interception methods mainly include three modes: Full pulse, partial pulse and interrupted sampling. Unconventional deceptive jamming is commonly generated by partial-pulse and interrupted-sampling modes, which include interrupted-sampling directly jamming (ISDJ), interrupted-sampling repeater jamming (ISRJ), interrupted-sampling circularly jamming (ISCJ), and partial-pulse dense transmitted jamming (PDTJ).

The generation mechanism of ISDJ is to use the DRFM technology to perform fast retransmission directly after intercepting part of the radar signal, and repeats the “interception-retransmission” process until it detects the falling edge of the radar signal. ISDJ is expressed as follows:(1)$$J_{ISDJ}\left(t\right)=\sum_{n=0}^{N_{J}-1}rect\left(\frac{t-\left(2n+1\right)T_{W}}{T_{W}}\right)s\left(t-T_{W}\right)$$
where s(t) stands for the intercepted radar signal, $N_{J}$ is the number of slices, and $T_{W}$ is the slice width.

The mechanism of ISRJ is similar to ISDJ, but every intercepted signal is retransmitted multiple times. The model of ISRJ can be written as:(2)$$J_{ISRJ}\left(t\right)=\sum_{m=1}^{M_{J}}\sum_{n=0}^{N_{J}-1}rect\left(\frac{t-\left(nM+n+m\right)T_{W}}{T_{W}}\right)s\left(t-mT_{W}\right)$$
where $M_{J}$ is the number of retransmissions.

ISCJ also belongs to interrupted-sampling modes. Not only is intercepted signal retransmitted, but the previously intercepted signal is also retransmitted in reverse order after primary sampling. ISCJ can be indicated as:(3)$$J_{ISCJ}\left(t\right)=\sum_{m=1}^{N_{J}}\sum_{n=1}^{N_{J}-m+1}rect\left(\frac{t-\alpha (m)-\beta (m,n)}{T_{W}}\right)s\left(t-\beta (m,n)\right),$$
where $\alpha \left(m\right)=\left[m\left(m-1\right)/2-1\right]T_{W}$ represents the time delay in intercepting the *m*-th slice, $\beta (m,n)=\left[n\left(n+1\right)/2+m\left(n-1\right)\right]T_{W}$ represents the time delay when the *m*-th slice is retransmitted for the *n*-th time.

PDTJ uses the interception method of the partial pulse. After intercepting a portion of the radar signal, it starts to continuously retransmit this sampled pulse, which is densely formed false targets directly on the radar range profile, without having to wait for the jammer to intercept and store the complete signal. PDTJ is expressed as follows:(4)$$J_{PDTJ}\left(t\right)=\sum_{m=1}^{M_{J}}rect\left(\frac{t-mT_{W}}{T_{W}}\right)s\left(t-mT_{W}\right)$$

### 2.2. Unconventional Barrage Jamming

According to different modulation modes, conventional barrage jamming can be divided into radio-frequency noise jamming, noise-amplitude modulation jamming, noise-frequency modulation jamming, etc. Their time domain models can be uniformly modeled as:(5)$$J_{CBJ}\left(t\right)=A_{J}\left(t\right)\exp\left\{j\left[\omega_{J}\left(t\right)\cdot t+\varphi_{J}\left(t\right)\right]\right\},$$
where $A_{J}\left(t\right)$, $\omega_{J}\left(t\right)$, and $\varphi_{J}\left(t\right)$ are the instantaneous amplitude, frequency and phase function of the jamming signal, respectively. Regardless of the modulation method, the contradiction between power and frequency band coverage cannot be solved.

In contrast, unconventional barrage jamming uses noise modulation on the intercepted radar signal to improve coherence with the radar signal, while preserving random and non-stationary jamming characteristics. It has the advantage of aiming at the radar carrier frequency with in-band high noise power. It can be mainly divided into noise productive jamming (NPJ) and noise convolutional jamming (NCJ).

NPJ is obtained by multiplying the intercepted radar signal by local noise in the time domain which can be derived as:(6)$$J_{NPJ}\left(t\right)=s\left(t\right)\times n\left(t\right),$$
where n(t) is the local noise. 

Local noise shifts the intercepted radar signal in the frequency domain, and the maximum spectral offset is the bandwidth of the local noise. When the radar signal is a pulse compression signal, since the jamming is related to the radar signal, jamming also obtains part of the gain through the matched filter. Therefore, the time domain of NPJ is similar to the noise, and its frequency domain automatically aligns with the center frequency of the radar signal, effectively reducing the power required for jamming under the premise of ensuring noise barrage.

Similarly, NCJ is to convolve the intercepted radar signal with local noise in the time domain, which can be written as:(7)$$J_{NCJ}\left(t\right)=s\left(t\right)⊗n\left(t\right).$$

NCJ is essentially the result of delay addition of the radar signal multiplied by different coefficients, so it can also obtain the matched filter gain, and barrage the signal in time domain with less power.

Sparse noise convolutional jamming (SNCJ) is an improvement based on NCJ, but combined with the interception method of interrupted-sampling. The intercepted radar signal is first sampled using a rectangular pulse series and then convolutionally modulated onto the local noise. Compare with NCJ, SNCJ can produce the same effect with less power. The model of the SNCJ is expressed as follows:(8)$$J_{SNCJ}\left(t\right)=\sum_{m=1}^{M_{J}}\sum_{n=0}^{N_{J}-1}rect\left(\frac{t-n\left(M+1\right)T_{W}-mT_{W}}{T_{W}}\right)s_{T}\left(t-mT_{W}\right)⊗n\left(t\right).$$

We can consider ISDJ as a special case of ISRJ when the number of retransmissions is one, and we can consider NCJ as a special case of SNCJ if the SNCJ sampling window is the entire radar signal. Therefore, this paper focuses on the recognition of radar unconventional active jamming, including ISRJ, ISCJ, PDTJ, NPJ, and SNCJ. 

## 3. Principle of Visibility Graphs 

### 3.1. Mathematic Principle

A graph is a system that contains a large number of individuals and interactions between individuals. If individuals are regarded as vertices, and interactions between individuals are regarded as a connection between vertices, then any complex system can be represented as a graph. 

A graph can be defined as a binary set, which is referred to as G={V,ε}. $V=\left\{v_{1},v_{2},\ldots ,v_{N}\right\}$ and $\varepsilon =\left\{e_{1},e_{2},\ldots ,e_{M}\right\}$, say, the vertex set and the edge set. The elements of V and ε are the vertices and edges, and the number of elements is denoted as order N=|V| and size M=|ε|. Each edge has a corresponding pair of vertices. The graph is called an undirected graph if vertex pair $\left(v_{i},v_{j}\right)$ and $\left(v_{j},v_{i}\right)$ corresponds to the same edge; otherwise, it is a directed graph, and $\left(v_{i},v_{j}\right)$ represents an edge from $v_{i}$ to $v_{j}$. Moreover, the graph is called a weighted graph if each edge is given a corresponding weight; otherwise, it is an unweighted graph.

The visibility algorithm converts a time series into a graph and is used for time-series analysis. Reference [15] applied the visibility algorithm and used the following definitions: Suppose $S=\left\{s_{i}\right\},i=1,2,\cdots ,N$ is a signal containing *N* sampled data, two arbitrary data $\left(a,s_{a}\right)$ and $\left(b,s_{b}\right)$ have visibility and consequently become two connected vertices $v_{a}$ and $v_{b}$ of the associated graph G, if any other data $\left(c,s_{c}\right)$ placed between them satisfy:(9)$$s_{c}<\left(s_{a}-s_{b}\right)\frac{b-c}{b-a}+s_{b}$$

The associated graphs derived from the visibility algorithm are called visibility graphs, and the number of vertices on visibility graphs is the same as the number of the data in a time series. For illustrative purposes, we plotted an example of the visibility algorithm in Figure 1. The upper part of Figure 1 is the first 20 data of a periodic time series; values are specifically 0.8, 0.5, 0.4, 0.6, 0.8, 0.5, 0.4, 0.6, 0.8, 0.5, 0.4, 0.6, 0.8, 0.5, 0.4, 0.6, 0.8, 0.5, 0.4, and 0.6. Each datum is represented as a column, and the height of the column represents the numerical value of the data. Here two data have visibility or invisibility if the tops of the corresponding two columns can or cannot be connected with a straight line. The lower part of Figure 1 shows the results of visibility graphs in a more concise form, and every vertex corresponds to series data in the same order on visibility graphs.

It is easy to verify that the visibility graphs obtained by the visibility algorithm have the following properties:Connectivity: Each vertex is connected to at least its left and right neighbors. If the vertex has only left (right) neighbors, it is at least connected to its left (right) neighbor;undirected and unprivileged: The generated network is an undirected and unprivileged network;affine transformation invariance: After rescaling both horizontal and vertical axes or after horizontal and vertical translations, the topology of the network does not change.

In complex-network theory, the degree of a vertex is defined as the number of connections the vertex has to other vertices, and degree distribution is then defined by the fraction of vertices in the network. It is noted that numerous real-world networks and theoretical networks satisfy certain degree distribution. Reference [15] found that the visibility algorithm can convert periodic time series into regular networks with Dirac degree distribution, which is the fingerprint of time-series periods. That means that visibility graphs can structurally conserve or inherit the regularity of periodic time series. Moreover, the visibility algorithm also converts random time series into random networks with exponential degree distribution and convert fractal time series into scale-free networks with power law degree distribution that can be utilized to detect the difference between random and chaotic series. 

Therefore, it is a natural idea that we can use the visibility algorithm to establish a natural bridge between the jamming signal and visibility graphs. The key question is to study the degree distribution characteristics that the signal may retain after being converted to visibility graphs.

### 3.2. Visibility Graphs for Analysis onSignals

#### 3.2.1. Sinusoidal Signal

In order to facilitate the degree-distribution characteristics of wideband-signal analysis, we first researched the sinusoidal signal on visibility graphs and assumed that a discrete sinusoidal signal is expressed as $s_{1}\left(t\right)=\cos\left(2\pi f_{0}t\right)$. We mainly discuss three regions in one period, since a sinusoidal signal is periodic, and defined them as $A:=\left(0,1/4f_{0}\right)$, $B:=\left(1/4f_{0},3/4f_{0}\right)$, and $C:=\left(3/4f_{0},1/f_{0}\right)$. Then, two data points that had visibility or invisibility with each other were defined to be written as $v_{i}↔v_{j}$ or $v_{i}↮v_{j}$. Any two data points on region *P* that had visibility or invisibility were defined to be written as P↔ or P↮.

For any two points $t_{B1},\;t_{B2}$ on region *B*, the following equations were satisfied:(10)$$\cos\left(2\pi f_{0}\left(\frac{t_{B1}+t_{B2}}{2}\right)\right)\in \left(-1,0\right),$$
(11)$$\cos\left(2\pi f_{0}\left(\frac{t_{B1}-t_{B2}}{2}\right)\right)\in \left(0,1\right).$$

Then we can find out that region *B* is a convex region:(12)$$\begin{array}{l} \frac{\cos\left(2\pi f_{0}t_{B1}\right)+\cos\left(2\pi f_{0}t_{B2}\right)}{2}\\ =\cos\left(2\pi f_{0}\left(\frac{t_{B1}+t_{B2}}{2}\right)\right)\cdot \cos\left(2\pi f_{0}\left(\frac{t_{B1}-t_{B2}}{2}\right)\right)\\ >\cos\left(2\pi f_{0}\left(\frac{t_{B1}+t_{B2}}{2}\right)\right) \end{array}$$

For λ∈[0,1], there is $s_{1}\left(\lambda t_{B1}+\left(1-\lambda \right)t_{B2}\right)<\lambda s_{1}\left(t_{B1}\right)+\left(1-\lambda \right)s_{1}\left(t_{B2}\right)$. Assume $t_{B3}=\lambda t_{B1}+\left(1-\lambda \right)t_{B2}$, and $s_{1}\left(t_{B3}\right)$ should meet: (13)$$\begin{array}{l} s_{1}\left(t_{B3}\right)<\lambda s_{1}\left(t_{B1}\right)+\left(1-\lambda \right)s_{1}\left(t_{B2}\right)\\ \;\;\;=\left(s_{1}\left(t_{B1}\right)-s_{1}\left(t_{B2}\right)\right)\frac{t_{B2}-\lambda t_{B1}-\left(1-\lambda \right)t_{B2}}{t_{B2}-t_{B1}}+s_{1}\left(t_{B2}\right)\\ \;\;\;=\left(s_{1}\left(t_{B1}\right)-s_{1}\left(t_{B2}\right)\right)\frac{t_{B2}-t_{B3}}{t_{B2}-t_{B1}}+s_{1}\left(t_{B2}\right) \end{array}$$

Therefore, any two points on region *B* always have visibility, that is, B↮. Similarly, regions *A* and *C* are concave regions; consequently, A↮ and C↮.

For any point on region *A*, pass data point $t_{A}$ for a tangent to $s_{1}\left(t\right)$ at data point $t_{LA}$. It is easy to find that, for any point $t_{M}$ on region *B* or region *C*, when $t_{M}<t_{LA}$ and $t_{A}<t_{P}<t_{M}$ are satisfied, then $\left[s_{1}\left(t_{M}\right)-s_{1}\left(t_{P}\right)\right]/\left(t_{M}-t_{P}\right)<\left[s_{1}\left(t_{M}\right)-s_{1}\left(t_{A}\right)\right]/\left(t_{M}-t_{A}\right)$, i.e., $t_{A}↮t_{M}$; when $t_{M}>t_{LA}$ and $t_{A}<t_{P}<t_{M}$ are satisfied, then $\left[s_{1}\left(t_{M}\right)-s_{1}\left(t_{P}\right)\right]/\left(t_{M}-t_{P}\right)>\left[s_{1}\left(t_{M}\right)-s_{1}\left(t_{A}\right)\right]/\left(t_{M}-t_{A}\right)$, i.e., $t_{A}↔t_{M}$. 

We define $t_{LA}$ as the left limit visible point (LVP) of $t_{A}$. Left or right LVP means it is the farthest point of all the points, which has visibility on the left or right side of $t_{A}$. $t_{LA}$ can be solved by tangential properties:(14)$$\frac{s_{1}\left(t_{LA}\right)-s_{1}\left(t_{A}\right)}{t_{LA}-t_{A}}=\frac{ds_{1}\left(t_{A}\right)}{dt},$$
(15)$$\cos\left(2\pi f_{0}t_{LA}\right)-\cos\left(2\pi f_{0}t_{A}\right)+2\pi f_{0}\left(t_{LA}-t_{A}\right)\sin\left(2\pi f_{0}t_{A}\right)=0.$$

Equation (15) is a transcendental equation. Its solution cannot be found analytically. For the $t_{A}$ on region *A*, the approximate solution was obtained by the graphical method. The change law of $t_{LA}$ with $t_{A}$ is shown in Figure 2a.

In addition, we should also find the right LVP $t_{RA}$ of $t_{A}$, which passes $t_{A}$ and is the tangent point of a tangent line on $s_{1}\left(t\right)$. It is easy to find that for any point $t_{N}$ on region *C* (because the right LVP must be in the region *C*), when $t_{N}<t_{RA}$ and $t_{A}<t_{Q}<t_{N}$, there is $t_{A}↔t_{N}$; when $t_{N}>t_{RA}$ and $t_{A}<t_{Q}<t_{N}$, there is $t_{A}↮t_{N}$.

Similarly, in order to get the solution of $t_{RA}$, the tangential property is available:(16)$$\frac{s_{1}\left(t_{RA}\right)-s_{1}\left(t_{A}\right)}{t_{RA}-t_{A}}=\frac{ds_{1}\left(t_{RA}\right)}{dt},$$
(17)$$\cos\left(2\pi f_{0}t_{RA}\right)-\cos\left(2\pi f_{0}t_{A}\right)+2\pi f_{0}\left(t_{RA}-t_{A}\right)\sin\left(2\pi f_{0}t_{RA}\right)=0.$$

Equation (17) is a transcendental equation and its analytical solution cannot be obtained. The approximate solution obtained by the graphical method was also used. The change law of $t_{RA}$ with $t_{A}$ is shown in Figure 2b. For normalization, the axes in the figure are multiplied by $f_{0}$.

In Figure 2a, as $t_{A}$ moves from the left end to the right end of region *A*, the position of $t_{LA}$ moves from the right end of region *C* to the left end of region *B*, and moving speed is gradually slowed down. When $t_{A}$ moves to the end of region *A*, the intersection of the tangent coincides with the vertex itself. In Figure 2b, as $t_{A}$ moves along the same route, $t_{RA}$ moves from the right end to the left of region *C*, but moving speed was almost negligible. Until $t_{A}$ stopped moving, $t_{RA}$ moved only about 3.5% (which we can see from the *Y*-axis of Figure 2b) of the total number of vertices in the entire period.

Then the visible region of $t_{A}$ is defined between $t_{LA}$ and $t_{RA}$, which means any point in the visible region has visibility with $t_{A}$. A more intuitive schematic diagram is shown in Figure 3. 

Similarly, for any point $t_{C}$ on region *C*, there is also a left LVP $t_{LC}$ on region *A*, and the right LVP $t_{RC}$ is on region *A* or *B*. The change laws of $t_{LC}$ and $t_{RC}$ are shown in Figure 4 where the change laws of $t_{LC}$ and $t_{RC}$ are mirrored with the change laws of $t_{RA}$ and $t_{LA}$.

Finally, consider any point $t_{B}$ on convex region *B*, and its left LVP $t_{LB}$ and right LVP $t_{RB}$ are on region *A* and region *C*, respectively. Similar to the derivation of $t_{A}$, we can obtain the change laws of $t_{LB}$ and $t_{RB}$ with $t_{B}$, which are shown in Figure 5. It is worth noting that $t_{LB}$, $t_{RB}$, $t_{RA}$, and $t_{LC}$, are tangent points, while $t_{LA}$ and $t_{RC}$ are not.

As can be seen from Figure 5, as $t_{B}$ moves from left to right on region *B*, $t_{LB}$ moves from right to left at a substantially constant speed in region *A*. When $t_{B}$ moves to the end, $t_{LB}$ does not reach the left end of region *A*. As $t_{B}$ moves from right to left on region *B*, the movement laws of $t_{RA}$ and $t_{LB}$ mirror each other.

It is obvious that the point on region *C* and the point on region *A* of the next period have invisibility (because $t=1/f_{0}$ has the largest data value) and do not add a new degree. Following the above discussion, we can solve degree $d_{1}$ of visibility graphs $G_{1}$ corresponding to $s_{1}\left(t\right)$:(18)$$d_{1}=\left\{ \begin{array}{l} f_{s}\left(t_{RA}-t_{LA}\right),\;region\;A\\ f_{s}\left(t_{RB}-t_{LB}\right),\;region\;B\\ f_{s}\left(t_{RC}-t_{LC}\right),\;region\;C \end{array}\right.,$$
where $f_{s}$ represents the sampling frequency. 

Assuming that $s_{1}\left(t\right)$ owns 300 data, which corresponds to three periods. The degree and its distribution of $G_{1}$ are shown in Figure 6.

As shown in Figure 6, the degree of vertices on region *B* change slightly, so the peak of the degree distribution is mainly related to the number of vertices on region *B*, and the peak position is $f_{s}\left(t_{RB}-t_{LB}\right)\approx 3f_{s}/4f_{0}$. Near the peak position, there is about a 50% probability in total because region B accounts for half of the entire period. Therefore, the sinusoidal signal $s_{1}\left(t\right)$ can be converted into a regular graph by the visibility algorithm, and the degree distribution of $G_{1}$ can be roughly expressed as single-spike distribution, which is given by:(19)$$P_{1}\left(d\right)=0.5\times \delta \left(d-\frac{3f_{s}}{4f_{0}}\right)$$

#### 3.2.2. Linear Frequency Modulation Signal

Linear frequency modulation (LFM) signal is a typical radar wideband signal. Suppose baseband LFM signal is $s_{2}\left(t\right)=\cos\left(\pi \mu t^{2}\right)$ and we define that the maximum repetition period (MRP) is from $t_{m1}$ to $t_{m2}$, where $t_{m1}$ is the data point corresponding to the maximum value, and $t_{m2}$ is the data point corresponding to the next maximum value. Although $s_{2}\left(t\right)$ is a non-periodic signal, it can be divided into three regions, *X*, *Y*, and *Z*, in one MRP. Region *X* is defined from $t_{m1}$ to data point $t_{01}$ corresponding to the first zero value, region *Y* is from $t_{01}$ to data point $t_{02}$ corresponding to the second zero value, and region *Z* is from $t_{02}$ to $t_{m2}$.

Similar to a sinusoidal signal, it is easy to prove that regions *X* and *Z* are concave regions, and region *Y* is a convex region, i.e., X↮, Y↔, Z↮. In three regions, degree $d_{2}$ of any vertex is also related to left LVP $t_{L}$ and right LVP $t_{R}$, as expressed by:(20)$$d_{2}=f_{s}\left(t_{R}-t_{L}\right).$$

Assuming that $s_{2}\left(t\right)$ owns 500 data corresponding to five MRPs. The degree and its distribution of $s_{2}\left(t\right)$ after being converted to visibility graphs $G_{2}$ by the visibility algorithm are shown in Figure 7. 

In Figure 7, every MRP has some vertices whose degrees are approximately equal, so degree distribution has a plurality of peaks with decreasing amplitude. Therefore, non-periodic baseband LFM signal $s_{2}\left(t\right)$ can still be converted to a regular network by the visibility algorithm, and degree distribution can be roughly expressed as multi-spike distribution as follows:(21)$$P_{2}\left(d\right)=\sum_{i=1}^{N_{R}}P_{i}\times \delta \left(d-d_{i}\right),$$
where $N_{R}$ is the number of MRPs, $P_{i}$ is the probability corresponding to the *i*-th MRP, $d_{i}$ is the degree corresponding to the *i*-th MRP.

Actually, the received signal includes not only the echo but also the noise signal obeying Gaussian distribution. In visibility graphs, it is easy to find that data with large values always have a large degree, and the degree of their adjacent data is relatively small. According to this property, the degree distribution of the Gaussian noise begins to exhibit Poisson distribution, but because of the existence of some vertices with large degree values (called hubs), the tail of the degree distribution gradually satisfies the exponential distribution. Therefore, Gaussian noise can be converted to an exponential random network by the visibility algorithm, whose degree distribution is different from sinusoidal and LFM signals which proves that degree distribution can distinguish radar signals and noises. Figure 8 illustrates the degree and its distribution of Gaussian noise with 300 data after converted to visibility graphs $G_{3}$.

In additive Gaussian noise environment, the actual received signal is randomly increased or decreased by a certain amount based on the original signal, which can be converted into the problem of the change of time-series local values. For a datum on the convex or concave region, the change (increase or decrease) of a local value causes a degree change of more than two vertices, indicating that network-structure changes amplify time-series changes to some extent. Hence, features on visibility graphs are more sensitive to noise than other features on the time or frequency domain.

We first consider the LFM signal with signal-to-noise ratio (SNR) is 10 dB, which owns 500 data. The degree and its distribution are shown in Figure 9.

Comparing Figure 9 with Figure 7, it can be seen that the degree of each vertex changed significantly after the introduction of Gaussian noise. The head and tail of degree distribution still satisfy Poisson distribution and exponential distribution which meet the degree distribution of Gaussian noise. The middle of the degree distribution had a spike, just like the LFM signal. Accordingly, we can understand the degree distribution of the actual LFM signal as a transition from multi-spike distribution to the exponential distribution.

#### 3.2.3. Jamming Signal

ISRJ, ISCJ, and PDTJ have a different number of MRPs while NPJ and SNCJ have noise characteristics similar to Gaussian noise in the time domain. That means that the degree distribution of different unconventional active jamming should have different characteristics owing to the above analysis. Because of the limitation of written space, we only show the degree distribution of ISRJ and NPJ in Figure 10 to verify the differences between the jamming.

Therefore, we have the following conclusions: The degree distribution of the sinusoidal and LFM signals satisfies single-spike distribution and multi-spike distribution, respectively, and the degree distribution of Gaussian noise is an exponential distribution. Moreover, the degree distribution of the actual received signal is between the ideal signal and the noise signal. Most importantly, different types of unconventional active jamming have different degree distributions on visibility graphs, which prepares for extracting more features (not only degree features) on visibility graphs, as shown below. 

## 4. Feature Extraction on Visibility Graphs

An adjacency matrix is a matrix used to characterize a graph adjacency relation. For a given graph G of order N, its adjacency matrix A is an N×N square matrix. If $v_{i}$ and $v_{j}$ on graph G have connected edges, the corresponding elements ${\left(A\right)}_{ij}$ of matrix A represent the weights of the edges; otherwise, by definition, ${\left(A\right)}_{ij}=0$. Because visibility graphs are unweighted, ${\left(A\right)}_{ij}=1$ if $v_{i}$ and $v_{j}$ on G have edges. According to this definition method, the adjacency of vertices on the graph can be completely represented by its adjacency matrix.

### 4.1. Average Degree

In addition to degree distribution, the average degree is another simple and important concept that describes the properties of the whole network. It averages the degree of all vertices in the graph to get the average degree $\bar{d}$, and is shown as:(22)$$\bar{d}=\frac{1}{N}\sum_{i=1}^{N}d_{i},$$
where $d_{i}$ is the degree of $v_{i}$. In addition, visibility graphs are undirected and unweighted, and the sum of the elements of the row or column of the adjacency matrix is also called the degree, so that the average degree is the sum of the diagonal elements of $A^{2}$:(23)$$\bar{d}=\frac{tr\left(A^{2}\right)}{N},$$
where $tr\left(A^{2}\right)$ represent the trace of matrix $A^{2}$.

The average degree can reflect the stability of the signal, and the larger the average degree is, the more stable the signal, since Gaussian noise can cause a sharp drop at the end of the degree distribution.

### 4.2. Average Clustering Coefficient

There is a phenomenon that, if a vertex has multiple vertices directly connected to it, then these neighbor vertices may also be directly connected to each other. This feature that is used to represent the clustering situation of vertices in the network is the clustering coefficient [23]. The clustering coefficient $C_{i}$ of a vertex $v_{i}$ is defined as:(24)$$C_{i}=\frac{2e_{i}}{d_{i}\left(d_{i}-1\right)},$$
where $e_{i}$ reflects the number of edges. When considering the undirected and unweighted graphs, the clustering coefficient of their vertices can also be obtained from: (25)$$C_{i}=\frac{{\left(A^{3}\right)}_{ii}}{{\left(A^{2}\right)}_{ii}\left[{\left(A^{2}\right)}_{ii}-1\right]}.$$

Averaging the clustering coefficients of all vertices, we obtain the average clustering coefficient of the network:(26)$$\bar{C}=\frac{1}{N}\sum_{i=1}^{N}C_{i},$$
obviously, $0\leq \bar{C}\leq 1$. When $\bar{C}=0$, all vertices in the network are isolated vertices, and no edges are connected; when $\bar{C}=1$, the network is a complete graph, that is, all vertices have edge connections between them. For a completely random network, $\bar{C}\to O\left(1/N\right)$ when the number of vertices is large enough. For jamming signals, the denser the vertices on visibility graphs are, the larger the average clustering coefficient.

### 4.3. Newman Assortativity Coefficient

In addition to the completely random network, there is always a correlation between different vertices, and degree distribution does not fully describe the characteristics of the network. In social networks, vertices tend to be connected with other vertices with similar degree values, which is referred to as assortativity; on the other hand, in biological networks, high degree vertices tend to attach to low degree vertices, referred to as disassortativity.

The assortativity coefficient calculation method based on the Pearson correlation coefficient was proposed by Newman [24]. This idea assumes that two vertices can be found by any one edge, and then two degrees are obtained. All edges are traversed, so that two sequences are obtained, and the Pearson correlation of the two sequences is analyzed, which is specifically defined as:(27)$$r=\frac{M^{-1}{\sum_{e_{ij}\in \varepsilon }d_{i}d_{j}-\left[M^{-1}\sum_{e_{ij}\in \varepsilon }\frac{1}{2}\left(d_{i}+d_{j}\right)\right]}^{2}}{M^{-1}\sum_{e_{ij}\in \varepsilon }\frac{1}{2}\left(d_{i}^{2}+d_{j}^{2}\right)-{\left[M^{-1}\sum_{e_{ij}\in \varepsilon }\frac{1}{2}\left(d_{i}+d_{j}\right)\right]}^{2}},$$
where M represents the total number of edges of the network. Obviously, 0≤|r|≤1. When r>0, the network is perfectly assortative; when r<0, the network is completely disassortative; when *r* = 0, the network is non-assortative. Therefore, the Newman assortativity coefficient can reflect the autocorrelation of data after the signal is converted to visibility graphs to some extent.

### 4.4. Normalized Network-Structure Entropy

Entropy is a measure used to describe the degree of disorder in a system and is often used in thermodynamics to characterize the state of a substance. In random networks, the importance of each vertex is equivalent, and its structure is considered to be disordered and has large entropy; in scale-free networks, there are a small number of vertices with large degree values (called hubs) and many vertices with small degree values, that is, its degree-distribution curve tends to decrease. Therefore, its structure is considered to be ordered and has small entropy.

Network-structure entropy is used to more succinctly measure the order state of complex networks. It is defined as:(28)$$E=-\sum_{i=1}^{N}I_{i}\cdot \ln I_{i},$$
where $I_{i}$ indicates the importance of vertex $v_{i}$.

It is easy to prove that a homogeneous network ($I_{i}=1/N$) has maximal entropy $E_{\max}=\ln N$; for a completely inhomogeneous network ($I_{1}=1/2$, $I_{i}=1/\left[2\left(N-1\right)\right]\left(i>1\right)$), the network has minimal entropy. Therefore, in order to eliminate the influence of number of vertices N on the entropy of the network structure, we normalize it to obtain normalized network-structure entropy $E_{norm}$:(29)$$E_{norm}=\frac{E-E_{\min}}{E_{\max}-E_{\min}}=\frac{-2\sum_{i=1}^{N}I_{i}\cdot \ln I_{i}-\ln4\left(N-1\right)}{2\ln N-\ln4\left(N-1\right)}.$$

## 5. Simulation and Discussion

In order to verify the effectiveness of the above features on visibility graphs, simulation experiments were performed with the parameters of the radar signal and the jamming signal, shown in Table 1 and Table 2.

After the received signal is converted to visibility graphs: (a) Average degree; (b) average clustering coefficient; (c) Newman assortativity coefficient; and (d) normalized network-structure entropy is selected to perform unconventional-active-jamming recognition. JNR was set in steps of 1 dB, and 100 Monte Carlo simulations were performed in the range of 0–25 dB. Figure 11 shows the values of these features, respectively. From Figure 11a, it can be seen that the average degree of unconventional barrage jamming (SNCJ, NPJ) was almost independent (always near 6) of the JNR, and the average degree of unconventional deceptive jamming (ISRJ, ISCJ, and PDTJ) increased with the increase of the JNR. Each jamming showed good separability when the JNR was greater than 10dB, and separability was greater when the JNR increases. In fact, in order to exhibit dense false-target characteristics after matching the filter, the JNR of the above-mentioned jamming was always more than 10 dB, so the average degree could be used well to distinguish unconventional active jamming.

Figure 11b shows that the average clustering coefficient of ISRJ increased with the increase of JNR while PDTJ decreased, and the average clustering coefficient of ISCJ was little affected by JNR. Through numerical analysis of the Newman assortativity coefficient of Figure 11c, it can be seen that the Newman assortativity coefficient of unconventional barrage jamming was always stable and SNCJ could be distinguished from NPJ. Moreover, unconventional barrage jamming on visibility graphs is assortative, since it has a positive Newman assortativity coefficient. Therefore, unconventional deceptive jamming can be distinguished through the average clustering coefficient and the Newman assortativity coefficient can be used as a visibility-graph feature of unconventional barrage jamming recognition. 

As can be seen from Figure 11d that, after conversion to visibility graphs by the visibility algorithm, all jamming signals had very large normalized network-structure entropy. We can analyze unconventional deceptive jamming and barrage jamming separately: Unconventional deceptive jamming can be well-distinguished under the condition that JNR is greater than 10 dB through normalized network-structure entropy; unconventional barrage jamming changes little with JNR and is always distinguishable. Therefore, normalized network-structure entropy is a suitable feature for radar unconventional-active-jamming recognition.

Random forests (RF) is a machine-learning method proposed by Leo Breiman and Adele Cutler for classification [25]. RF combines decision-tree classifiers with the Bagging algorithm, specific for using each sample subset obtained by the Bagging algorithm to construct the decision trees. The significance of constructing the RF classifier is to randomly generate multiple decision trees, and each decision tree does not need to have high classification accuracy (a weak classifier). At the same time, by using the Bagging algorithm to combine multiple decision trees, the over-fitting problem is well-solved, and overall generalization ability is improved.

Therefore, we realized a radar unconventional-active-jamming recognition scheme by using the RF classifier, and an RF structure diagram was illustrated in Figure 12. The radar unconventional-active-jamming recognition method based on the RF algorithm is summarized as follows:Converting unconventional active jamming from the time domain to visibility graphs using the visibility algorithm;Extracting the average degree, average clustering coefficient, Newman assortativity coefficient, and normalized network-structure entropy as four-dimensional features;Applying the Bagging algorithm to extract training samples from four-dimensional features and corresponding jamming category labels. Training decision trees until the number of decision trees reaches the preset threshold;Generating RF structures according to Figure 12a;Sending the test samples to the RF classifier, and the classification results of all decision trees are voted according to Figure 12b. The category of jamming with the largest number of votes is taken as the final output of the algorithm.

In the RF classifier, we can further improve recognition performance by (a) adding number of decision trees $N_{DT}$, (b) increasing maximum depth of decision trees $D_{DT}$, (c) adding number of splits $N_{S}$, and (d) changing the decision-tree splitting algorithm. Table 3 shows recognition performance based on the RF classifier under different parameters. According to Table 3, we found that the performance of the RF classifier was hardly affected by adding the number of splits, and could be improved well by adding the number of decision trees. Although the computational time of the algorithm increases rapidly with the deepening of the decision trees, it obviously improves the recognition probability under a different JNR, especially at a low JNR. Moreover, the ID3 algorithm is always used as the splitting algorithm of decision trees because other algorithms negligibly improve for small training samples and need the extra computational time. Therefore, in the actual situation, we prefer to choose a minor number of decision trees and splits, and appropriate decision-tree depth to ensure a balance between recognition probability and computational time.

We chose Groups 6 and 7 as the ultimate simulation parameters. The recognition probability of unconventional active jamming with a JNR setting from 0 to 25 dB, in 1 dB steps, is shown in Figure 13. In Figure 13a,b it is shown that, when the JNR was higher than 0 dB, the average recognition probability of the algorithm was over 90%. In Figure 13b, we can see that the recognition probability for each type of jamming was always higher than 95% when the JNR was 5 dB or higher. This simulation proves that the unconventional-active-jamming recognition method proposed in this paper is robust to the presence of noise in the jamming, and effective in identifying the type of jamming after the JNR is higher than 5 dB, which can often be satisfied.

## 6. Conclusions

In this paper, a visibility-graph based unconventional-active-jamming recognition scheme is proposed. Theoretical analyses first showed that different types of jamming exhibit different degree distributions on visibility graphs. Then, other valid features on visibility graphs were calculated: Average degree, average clustering coefficient, Newman assortativity coefficient, and normalized network-structure entropy. Finally, an RF classifier based on these four features was proposed to achieve the recognition of unconventional active jamming. Numerical simulations demonstrated that average recognition probability is always greater than 97%, regardless of JNR, by reasonably selecting the classifier parameters. Further studies will be carried out on the combination of different types of jamming.

## Figures and Tables

**Figure 1 sensors-19-02344-f001:**
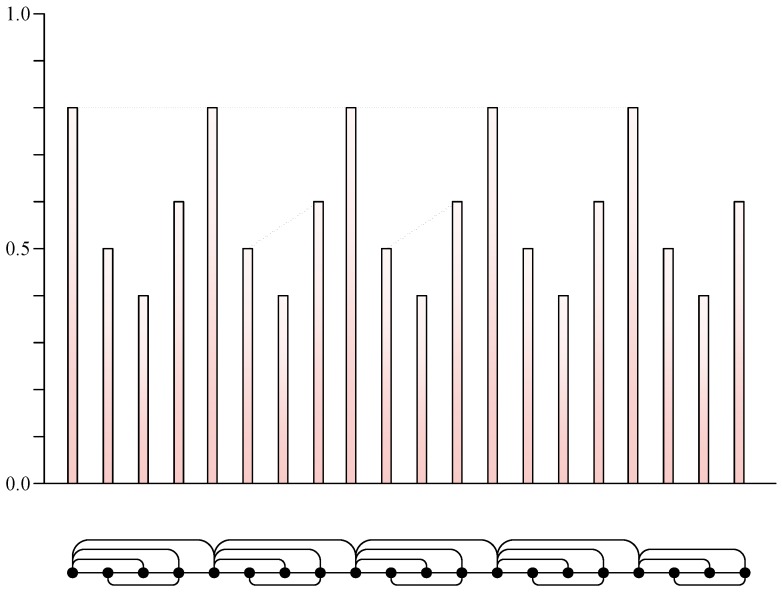
Schematic diagram of visibility graphs.

**Figure 2 sensors-19-02344-f002:**
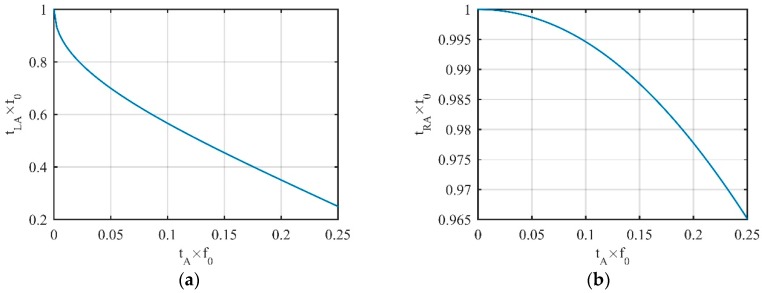
Change laws of (**a**) $t_{LA}$ and (**b**) $t_{RA}$ with $t_{A}$.

**Figure 3 sensors-19-02344-f003:**
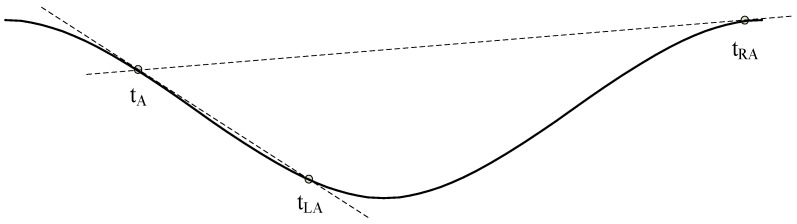
The visible region of $t_{A}$.

**Figure 4 sensors-19-02344-f004:**
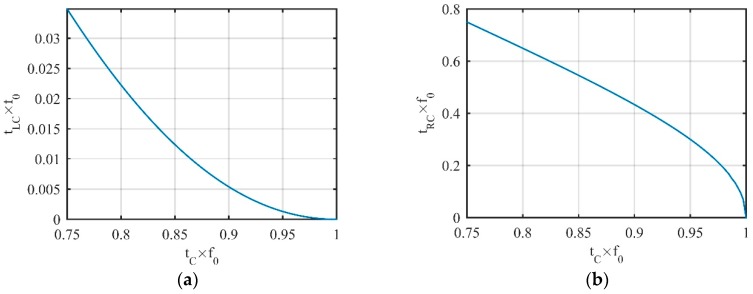
Change laws of (**a**) $t_{LC}$ and (**b**) $t_{RC}$ with $t_{C}$.

**Figure 5 sensors-19-02344-f005:**
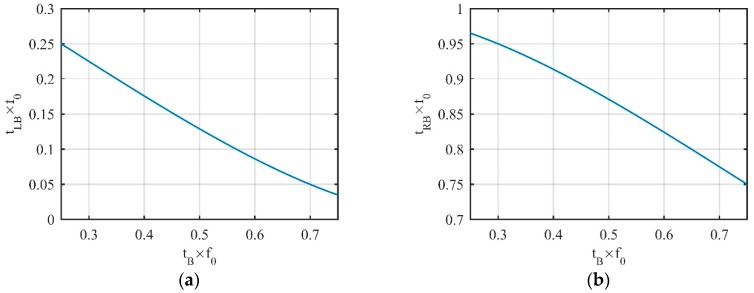
Change laws of (**a**) $t_{LB}$ and (**b**) $t_{RB}$ with $t_{B}$.

**Figure 6 sensors-19-02344-f006:**
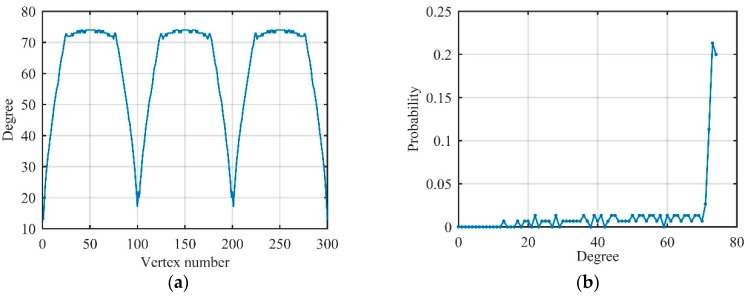
Related degree features of the sinusoidal signal on visibility graphs. (**a**) Degree of each vertex; (**b**) degree distribution.

**Figure 7 sensors-19-02344-f007:**
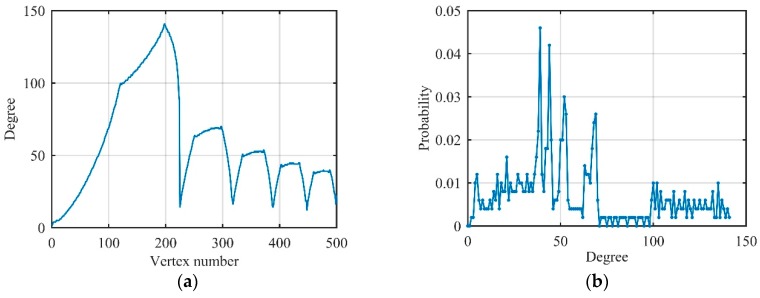
Related degree features of linear frequency modulation (LFM) signal on visibility graphs. (**a**) Degree of each vertex; (**b**) degree distribution.

**Figure 8 sensors-19-02344-f008:**
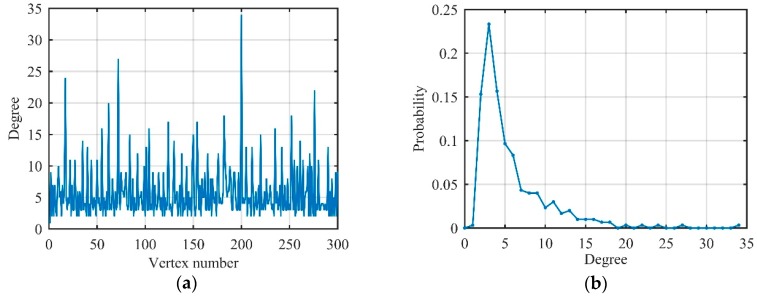
Related degree features of Gaussian noise on visibility graphs. (**a**) Degree of each vertex; (**b**) degree distribution.

**Figure 9 sensors-19-02344-f009:**
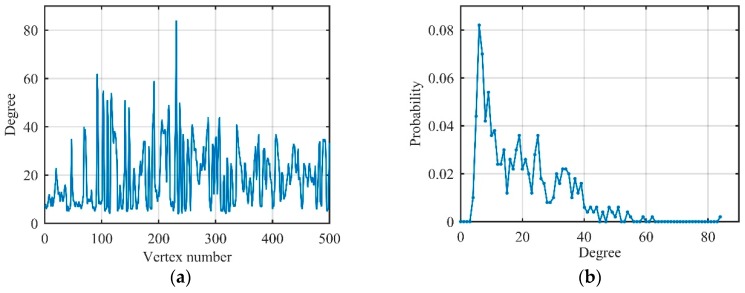
Related degree features of LFM signal on visibility graphs (signal-to-noise ratio (SNR) = 10 dB). (**a**) Degree of each vertex; (**b**) degree distribution.

**Figure 10 sensors-19-02344-f010:**
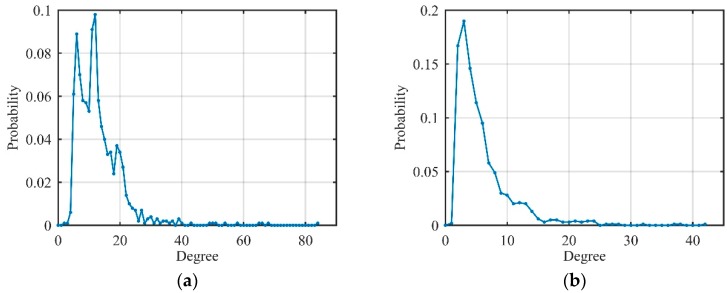
Degree distribution of jamming signals on visibility graphs (jamming-to-noise ratio (JNR) = 10 dB). (**a**) Interrupted-sampling repeater jamming (ISRJ); (**b**) noise productive jamming (NPJ).

**Figure 11 sensors-19-02344-f011:**
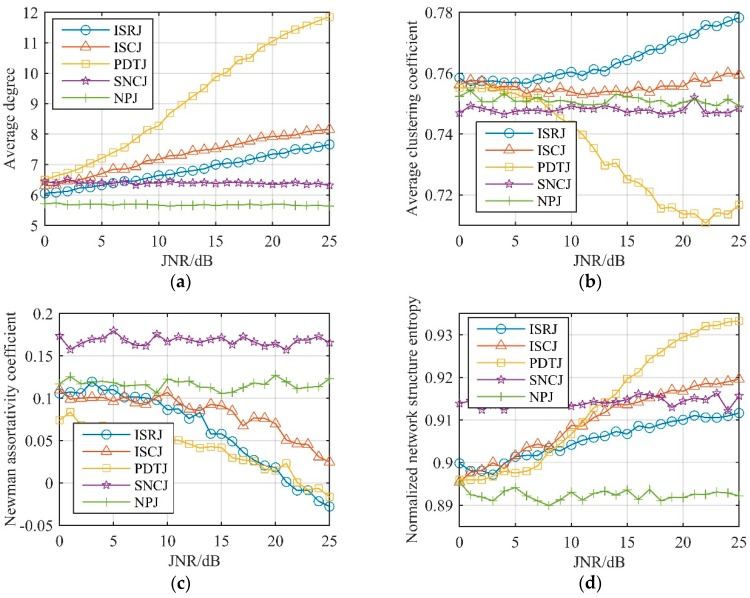
Related features of unconventional active jamming on visibility graphs. (**a**) Average degree; (**b**) average clustering coefficient; (**c**) Newman assortativity coefficient; (**d**) normalized network-structure entropy.

**Figure 12 sensors-19-02344-f012:**
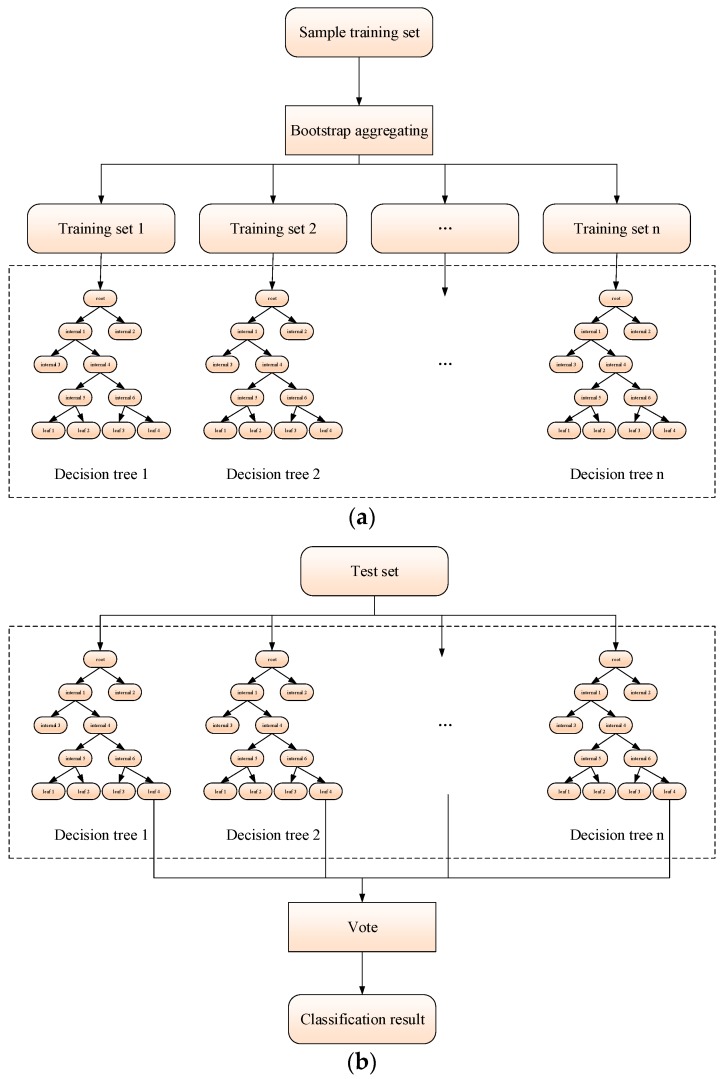
Random-forests (RF) structure diagram. (**a**) Training process; (**b**) test process.

**Figure 13 sensors-19-02344-f013:**
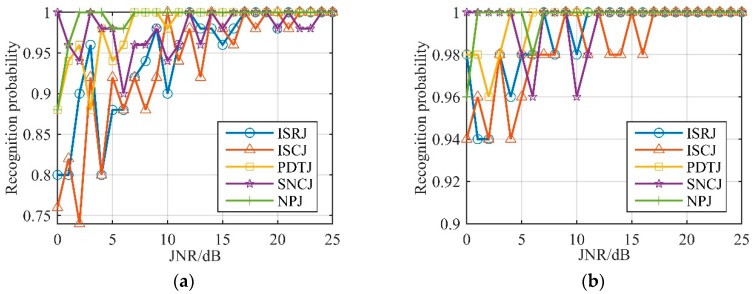
Recognition probability of unconventional active jamming. (**a**) Group 6; (**b**) Group 7.

**Table 1 sensors-19-02344-t001:** Simulation parameters for the radar signal.

Signal System	Sampling Frequency	Signal Bandwidth	Receiver Bandwidth	Pulse Width	Pulse Repetition Interval
LFM signal	50 MHz	5 MHz	10 MHz	20 μs	200 μs

**Table 2 sensors-19-02344-t002:** Simulation parameters for the jamming signal. Note: ISCJ, interrupted-sampling circularly jamming; PDTJ, partial-pulse dense transmitted jamming; SNCJ, sparse noise convolution jamming.

Jamming Type	Parameter and Value
ISRJ	Slice width, 1 μs; retransmitted count, 3
ISCJ	Slice width, 1 μs
PDTJ	Slice width, 4 μs
SNCJ	Gaussian white noise; slice width, 4 μs; noise bandwidth, 10 MHz
NPJ	Gaussian white noise; pulse width, 20 μs; noise bandwidth, 10 MHz

**Table 3 sensors-19-02344-t003:** Recognition performance based on the RF classifier. Note: ARP, average recognition probability; CT, computational time.

Group	Parameter			Result				
	N_DT_	D_DT_	N_S_	ARP (JNR = 0 dB)	ARP (JNR = 5 dB)	ARP (JNR = 10 dB)	ARP (JNR = 15 dB)	CT
1	20	2	2	66.8%	72.8%	76.4%	86.0%	0.25 s
2	40	2	2	68.4%	74.4%	83.6%	89.2%	0.51 s
3	80	2	2	69.6%	76.8%	87.6%	95.2%	0.92 s
4	160	2	2	72.4%	78.0%	88.0%	96.0%	1.91 s
5	20	4	2	76.0%	84.0%	92.8%	95.6%	1.11 s
6	20	8	2	90.0%	93.2%	97.2%	99.2%	13.57 s
7	20	12	2	97.2%	98.0%	98.8%	100.0%	125.22 s
8	20	2	4	66.0%	72.4%	77.2%	87.2%	0.30 s
9	20	2	8	67.6%	70.0%	78.4%	87.5%	0.42 s
10	20	2	16	64.4%	70.4%	79.2%	88.0%	0.63 s

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
