# Peer review of "Novel Unconventional-Active-Jamming Recognition Method for Wideband Radars Based on Visibility Graphs"

_sensors, 2019, doi:10.3390/s19102344_

Round 1

Reviewer 1 Report

The paper attempts to use graph theoretic formulation to detect jamming in a wideband radar. While the premise is interesting, the article is very poorly written and coins its own new terms at the turn of pretty much every paragraph without defining them. The abstract itself is written as bullet points and is reflective of poor writing throughout the article.

The novelty of the technique remains questionable because the ideas are presented very incoherently. The core of the article, i.e., Section 3, jumps from one idea to the other and, in the end, it remains unclear how graph-theoretic formulation has helped in solving the problem. The Introduction is overly long and included write-ups on totally unconnected theory.

There are no comparisons with state-of-the-art to show that the proposed method is better.

Author Response

The authors sincerely thank all anonymous reviewers for their insight peer review comments. And we deeply appreciate editors for their patient helps.

Point 1: While the premise is interesting, the article is very poorly written and coins its own new terms at the turn of pretty much every paragraph without defining them.

Response 1: Every new term has been defined after the first appearance. “The visibility algorithm” is defined in line 150, “visibility graphs” is defined in line 156, “visibility” and “invisibility” are defined in line 162, “degree distribution” is defined in line 176 and so on. They are corrected in the revised version of the paper.

Point 2: The abstract itself is written as bullet points and is reflective of poor writing throughout the article.

Response 2: Thank you for point out this problem. Therefore, we decide to rewrite the abstract in order to reduce the doubts when readers read the text.

Point 3: The novelty of the technique remains questionable because the ideas are presented very incoherently.

Response 3: The core idea of this paper is mining the differences between jamming and echo, and visibility graphs feature can amplify these differences. Section 3 and 4 show the rationality and effectiveness of visibility graphs feature. And in order to make the logic more coherent, we modify certain content.

Point 4: It remains unclear how graph-theoretic formulation has helped in solving the problem.

Response 4: There are several methods of graph transformation and the visibility algorithm is one of them. Until now, the essence of the transformation method is still a problem waiting to be solved, that is also our aim in the next stage.

Point 5: The Introduction is overly long and included write-ups on totally unconnected theory.

Response 5: We almost rewrite the introduction. They are corrected in the revised version of the paper.

Point 6: There are no comparisons with state-of-the-art to show that the proposed method is better.

Response 6: Most recognition methods are mainly for conventional jamming recognition, such as RGPO/VGPO/RVGPO. It is hard to compare performance differences.

Reviewer 2 Report

The work considered an important and necessary scientific issue.

There is a lack of citation in the text of the work listed at the end of the literature.

There is also a lack of literature references to mathematical problems, which makes it difficult for the reviewer to evaluate the innovativeness of these mathematical formulas.

The interpretation of the results obtained should be supplemented (extended).

Author Response

The authors sincerely thank all anonymous reviewers for their insight peer review comments. And we deeply appreciate editors for their patient helps.

Point 1: There is a lack of citation in the text of the work listed at the end of the literature.

Response 1: We may be the first to put forward the application of network science to electronic counter-countermeasures, so we mainly cite some classic literature about these two fields. Nevertheless, we add some citation.

Point 2: There is also a lack of literature references to mathematical problems, which makes it difficult for the reviewer to evaluate the innovativeness of these mathematical formulas.

Response 2: We add some literature references to mathematical problems. Reference [15] gives the visibility algorithm for time series analysis. Reference [20] defines the average clustering coefficient. Reference [22] proposes the Newman assortativity coefficient. And Reference [23] introduces random forests algorithm for classification. Moreover, proof in Section 3.2 is original.

Point 3: The interpretation of the results obtained should be supplemented (extended).

Response 3: Thank you for point out this problem. We extend some interpretation after each figure and rewrite the conclusions. They are corrected in the revised version of the paper.

Reviewer 3 Report

line 18

in the abstract the term "Degree distribution" is not clear to Readers that are not active Researchers in the field. May be you add one sentence, what this metric is about.

line 74

"A series of Achievements …"  drop "There are"

line 78

"Section 3 first introduces…"

add: "method for the received signal;"

line 80

2x "based", please rephrase

line 91

"...to perform fast retransmission directly…"

line 97

"where s(t) stands! for…"

line 100

"...can be expressed as:"

line 108

"...and retransmission strategies…"

line 109

"...a slice of the radar signal…"

line 115

"...can be expressed as:"

line 118

"ISCJ also belongs to an …"

line 122

"m-th slice"

line 123

"...and the n-th retransmission."

line 138-140

sentence confusing "Among them...to supress jamming"

argumentation is unlogical.

line 143

"...can achieve full coverage of frequency agilitry" phrasing is confusing. What do you want to say?

line 144

"jamming power spectral density" PSD in [W/Hz] - is this what you want to say?

line 206

"Besides, each..." add comma

line 215

"...and uses the following definitions:"

line 219

People not familiar with VG cannot follow you here.

The graph Fig 1  comes too abrupt with no chance to understand how this is constructed.

line 243

"Degree Distribution" - add a sentence what this distribution indicates.

line 247

"The two verstices ...with each other. They are written as:" split into two sentences!

line 248

"not visibility" this is unclear to the readers. What does it mean?

line 253

"Then get the Region B..." sentence is not proper constructed.

line 265

"can be solved" passive voice

line 268

"Eq. 19 is a transcendental equation. Its solution cannot be found analytically."

break into two sentences.

line 280

"Eq. 21 is a transcendental equation and its analytical..."

line 294

"Until the end..." sentence not properly phrased. Confusing

line 305

"...while at point t_A, only …"

line 317

"Assume that the simple pulse signal interepts…" drop <which>

line 318

"after transformation to the"

line 326

"...expressed as the Dirac Distribution" The Dirac isn't a function, it is a distribution.

line 341

"are shown"

line 345

"outside the peaks" plural

line 350

"is the Degree corresponding…" drop <of>

i-th

line 358

"having a large value"

line 393

"is a single-spike"

"is a multi-spike distribution"

line 402

"If the vertices v_i and v_j on the graph..."

line 403

"...otherwise by Definition (A)_ij=0."

line 407

"unconventional jamming" - is this a settled term in literature? If not, please explain why you use this term and from what other kind of jamming this stands apart.

line 417

"is also called the degree"

line 434

"...reflects the number of edges, that actually .."

line 435

"in Addition..." sentence construction is confusing

line 453

"first defines"

line 463

"which is specifically defined"

line 467

"the network is perfectly assortative"

line 474

"In the Network..." sentence construction confusing.

line 479

"...succinctly. It is defined as:" break into two sentences

line 499

"After...by VG algorithm, a)average Degree, b)average clustering coefficient…" - and so on, no slashes "/"

line 511

"...always is more that.."

line 530

"...non-homgenity"

line 533

"...a high Degree of distinguish" - sentence construction confusing. Not clear what is meant.

line 537

"...and uses each sample.."

line 573

"...from Figure 13, when assuming the chirp System"

Thank your for excellent work.

Reviewer

Author Response

The authors sincerely thank all anonymous reviewers for their insight peer review comments. And we deeply appreciate editors for their patient helps.

Point 1: In the abstract the term "degree distribution" is not clear to readers that are not active researchers in the field. Maybe you add one sentence, what this metric is about.

Response 1: Although "degree distribution" is a common concept in network science, it may not clear to researchers in the electronic counter-countermeasures field. In order to reduce the doubts when readers read the text, we decide to change the writing form of the abstract. They are corrected in the revised version of the paper.

Point 2: Some mistakes with English expression.

Response 2: Really thank you for patiently point out these problems. They are all corrected in the revised version of the paper.

Point 3: Some confusing sentences.

Response 3: Really thank you for patiently point out these problems. We rewrite the introduction, conclusions and some confusing paragraphs. They are more clearly expressed in the revised version of the paper.

Round 2

Reviewer 1 Report

This is a revision of an earlier manuscript. While the writing has improved from last time, there remain significant English language issues. The authors should consult a professional editor. The writing is very bad at too many places to list here.

The paper attempts to model the radar time-series data as a visibility graph. It is unclear why this is a good idea. In practice, the time series data contain thousands of samples. Analyzing them as a graph will lead to extremely high computational cost.

Many of the key ideas are presented in such a complicated way that it is very difficult to follow them. The core concept is to model the received signal as a graph and then compare its degree distribution with some reference. Presence of a jammer in the signal will distort such distribution and this can be inferred by feature extraction from the graph. While this is a nice concept, the presentation is very unclear for an uninitiated reader.

Further, the paper does not compare itself with standard state-of-the-art in jammer removal in terms of performance, complexity and computational time. Without such comparisons, the value of the proposed method cannot be ascertained.

Author Response

Response to Reviewer 1 Comments

The authors sincerely thank all anonymous reviewers for their insight peer review comments. And we deeply appreciate editors for their patient helps.

Point 1: This is a revision of an earlier manuscript. While the writing has improved from last time, there remain significant English language issues. The authors should consult a professional editor. The writing is very bad at too many places to list here.

Response 1: We have paid for MDPI English editing service, and modify many places. We are very sorry for English language issues.

Point 2: The paper attempts to model the radar time-series data as a visibility graph. It is unclear why this is a good idea. In practice, the time series data contain thousands of samples. Analyzing them as a graph will lead to extremely high computational cost.

Response 2: Figure 9 shows that network-structure changes amplify time-series changes to some extent, which is more advantageous for identifying jamming. We propose a new thought to signal analysis at high computational cost, and there are still many places worth exploring. Moreover, the fast visibility algorithm or other algorithms, such as recurrence networks, are the subject of the next study.

Point 3: While this is a nice concept, the presentation is very unclear for an uninitiated reader.

Response 3: We have modified many places in Section 1, 3, and 4, to make the presentation of this article clearer.

Point 4: Further, the paper does not compare itself with standard state-of-the-art in jammer removal in terms of performance, complexity and computational time. Without such comparisons, the value of the proposed method cannot be ascertained.

Response 4: The existing active-jamming-recognition method is mainly used for identifying RGPO, VGPO, and RVGPO. In Table 3, We compare the recognition performance horizontally by changing the classifier parameters.

Round 3

Reviewer 1 Report

In the revision, authors have done a good job of improving the manuscript. English languages issues have largely been overcome through professional editing.